# Tellurium Nanotubes and Chemical Analogues from Preparation to Applications: A Minor Review

**DOI:** 10.3390/nano12132151

**Published:** 2022-06-22

**Authors:** Cailing Liu, Ruibin Wang, Ye Zhang

**Affiliations:** School of Chemistry and Chemical Engineering, University of South China, Hengyang 421001, China; a15078350463@163.com

**Keywords:** tellurium nanotubes, preparation, growth mechanism, application

## Abstract

Tellurium (Te), the most metallic semiconductor, has been widely explored in recent decades owing to its fantastic properties such as a tunable bandgap, high carrier mobility, high thermal conductivity, and in-plane anisotropy. Many references have witnessed the rapid development of synthesizing diverse Te geometries with controllable shapes, sizes, and structures in different strategies. In all types of Te nanostructures, Te with one-dimensional (1D) hollow internal structures, especially nanotubes (NTs), have attracted extensive attention and been utilized in various fields of applications. Motivated by the structure-determined nature of Te NTs, we prepared a minor review about the emerging synthesis and nanostructure control of Te NTs, and the recent progress of research into Te NTs was summarized. Finally, we highlighted the challenges and further development for future applications of Te NTs.

## 1. Introduction

Tellurium (Te), as an intrinsic p-type semiconductor with a tunable bandgap ranging from 0.3 to 1.05 eV, has broad application prospects in the fabrication of many modern devices. Normally, Te has two allotropes, one is amorphous Te with a black color, and the other one is hexagonal crystalline Te with metallic properties in a silver–white color, which is the most stable morphology at normal temperature and pressure. Hexagonal Te is highly anisotropic and consists of helical chains of covalently bound atoms, which are in turn bound together by van der Waals interactions to form a hexagonal lattice, resulting in Te with pronounced chirality, also making Te readily grow unidirectionally so to form one-dimensional (1D) nanostructures such as Te nanowires (NWs), Te nanobelts (NBs), Te nanorods (NRs), and Te nanotubes (NTs).

The 1D nanostructured Te has drawn tremendous attention due to its intriguing properties, such as excellent thermoelectricity, high piezoelectricity, fast photoconductivity, nonlinear optical effects, and high sensitivity for gas (such as NO, NO_2_, and CO, etc.) and ions [1,2,3]. The successful construction of a 1D nanostructured Te counterpart to other 1D nanomaterials is strongly related to its size and shape. Size has an effect on its specific surface area and surface-to-volume atoms ratio, while the shape affects not only the facet size but also the content of surface atoms [4,5,6]. In this case, 1D Te nanomaterials include Te NWs, Te NBs, and Te NRs that are of relatively simple shapes and are outperformed by Te NTs. Different from the other three, the existence of the inner surface of Te NTs can help them to stay clean and intact during the fabrication of Te NT-based devices for enhanced performance. On the other hand, Te NTs of many shapes, such as cylindrical, prismatic (hexagonal column, trigonal column, etc.), bamboo-like, and shuttle-like, have been reported in recent years [7,8,9,10,11,12,13]. By tuning the Te-containing precursor, the surfactant, reaction temperature, reaction time, and reducing reagent, the aspect ratio, morphology, and architecture of Te NTs can be rationally designed [7,14,15]. To the best of our knowledge, the advantages of these Te NTs are varied in shapes, further integrating or chemically compositing with other elements/compounds, which can facilitate their applications in specific areas. For example, Te NTs with different hollow structures can be engineered to generate some tubular and nanowire-in-nanotube advanced functional materials, Bi_2_Te_3_ NTs, CdTe NTs, Te@Bi van der Waals heterojunctions, and carbon-coated MoS_1.5_Te_0.5_ nanocables [16,17,18,19], and have promising applications in gas/ion sensing, catalysis, photodetectors, and energy storage. These results are attributed to their high specific surface area, strong interpenetrating network, and good electron/ion transport.

Although increasing attention has been devoted to Te NTs, the preparation methods, physical/chemical properties, and application prospects of them have been only occasionally reviewed. In 2017, the synthesis and geometric tailoring of the nanostructured Te reported in recent decades were outlined by Zhen He et al. [6], where the characteristics and applications of those Te nanostructures of 0D, 1D, 2D, and 3D were comprehensively discussed. As the field of Te NTs is rapidly evolving, a periodic update focusing on their recent progress is necessary. Here, we emphasize the latest significant developments in their synthetic strategies (regarding the effects of reaction time, temperature, pH, and others), property/structural optimization, and emerging applications. Finally, the challenges and future applications of Te NTs are summarized and prospected, along with some instructive suggestions.

## 2. Synthesis of Te NTs

### 2.1. The Synthetic Strategies of Te NTs

Given that the synthetic strategies of Te NTs account for their shape differences, it is important to understand the related growth mechanisms. In recent years, cylindered Te NTs, prismatic Te NTs, bamboo-like Te NTs, and shuttle-like Te NTs have been reported (Figure 1). These Te NTs of different shapes, end-opened or both-ends opened, are mainly fabricated by liquid-phase methods and gas-phase methods. In general, there are Te NTs of only one dominant shape formed during either of these methods, which is strongly related to the conditions for the nucleation and growth of Te NTs. In other words, all of the reaction time [8], temperature [7,20], pH [14,21], the concentration of Te-containing precursor [7], sacrificial template [11,22,23], surfactant [9,12,15], reducing reagent [15,24], and others (organic additives [12], with or without photothermal assistance [25], and with or without microwave assistance [13]) play a role in the formation of Te NTs with different shapes and performance.

#### 2.1.1. Liquid-Phase Methods

Liquid-phase methods are most frequently used for the synthesis of Te NTs due to their low cost and ease of scale-up, which usually involve the polyol reflux method, the hydrothermal/solvothermal method, the template induction/sacrifice method, the electrochemical method, the photothermal assisted method, and the ultrasonic-/microwave-assisted method [7,14,22,25,27,28].

The polyol reflux method is a conventional liquid-phase method where high-performance Te NTs can be fabricated under the optimized temperature, reflux time, reflux solvent, and template [7,9,20]. In 2002, inspired by the polyol process developed by Fievet et al. to generate colloidal particles of metals, ethylene glycol, that could serve as both solvent and reducing reagent, was used by Mayers and Xia to grow single-crystalline, geometrically closed, concentric, and cylindrical Te NTs [7]. During the procedure, orthotelluric acid was added to pure ethylene glycol and refluxed at 197 °C, included in which the growth time and the initial concentration of orthotelluric acid were tuned for the optimal length and diameter of the Te NTs. In the same year, the trigonal monodispersed Te NTs were synthesized through a solution-phase, self-seeding process, also by them [20]. It was found that their geometry could be tailored by altering the refluxing temperature from below 100 °C (spines), 100–160 °C (filaments), 160–180 °C (needles), to above 180 °C (tubes).

As an alternative to the polyol reflux method, hydrothermal and solvothermal methods are also frequently used for the large-scale production of Te NTs due to their easily controllable reaction conditions (pH, organic additives, and reducing agents), especially considering the sensitive pH-responsiveness of Te NTs. In 2003, single-crystalline trigonal Te NTs with well-controlled shapes and sizes were formed by hydrothermally reducing Na_2_TeO_3_ in the ethanol/water mixture at 100 °C [14]. Their nucleation and growth rates were varied in the pH of the reaction media so that 1 M of HCl led to the formation of NWs, while 1 M of NaOH was instead favored for forming NTs. Besides the pH, organic additives, especially biological molecules with specific functional groups, could also affect the shape and size of Te NTs during the hydrothermal/solvothermal treatment. Chemically, they were surfactant-like compounds that could lead and orient the growth of crystalline Te. He et al. controlled the growth of shuttle-like Te NTs with sharp tips and amino acids [12]. To understand the effect of functional groups on their structural conformation and morphology, lysine, histidine, phenylalanine, cysteine, glycine, and serine were adopted. Their results indicated that serine could influence the surface tension to induce the selective formation of shuttle-like Te NTs with two sharp and flexible tails. Except for the additive, the type and concentration of reducing agents are also important for growing well-shaped Te NTs. Zhong et al. proposed a fabrication strategy of Te NTs under solvothermal conditions based on reductive ascorbic acid that could significantly affect the geometry of the Te nanostructures [15]. Te NTs were formed within the ascorbic acid range of 1.00–2.25 mL, while a higher addition produced Te NRs instead.

It has been demonstrated that metals, alloys, and metal oxide can induce or be sacrificed to generate Te NTs with a customizable length, diameter, and thickness, while sometimes the electrochemical treatments are needed. As shown in Figure 2, using cobalt NWs as the sacrificial template, the diameter and thickness of Te NTs were facilely adjusted at room temperature based on a galvanic displacement reaction [23]. The same process could also be performed on the segmental NiFe NWs to synthesize bamboo-like Te NTs, which were also similar to heterostructures [11]. Further investigations suggested that the segment length and stalk thickness of them were dependent on the length and varied Fe-rich/Ni-rich segments of the NiFe NWs template. Compared to metals and alloys, metal oxides are more stable against environmental moisture/oxygen and heat. Kapoor et al. recently reported an etching-free method to grow CdTe NTs on an anodic alumina template through the direct current electrodeposition [29]. Without the multiple etching steps, the uniform geometry of the template was inherited by these CdTe NTs to present a high aspect ratio, ordered pores, and a controllable diameter. Additionally, Te NTs could also be electrochemically produced by excluding the use of the template, which significantly decreased the cost, processing complexity, and environmental hazard. Based on a template-free electrodeposition method, polycrystalline and evenly elementally-distributed CdTe NTs, which included the (111), (220), and (311) crystal planes, were formed through the Kirkendall effect [30].

In addition, Te NTs were reported to be efficiently fabricated with the assistance of photothermal, ultrasonication, or microwave radiation. A facile and environmentally friendly visible-light-driven synthesis method of Te NTs was developed by Zhang et al [25]. In the presence of polyvinylpyrrolidone or polyvinyl alcohol, single-crystalline Te NTs, as well as NWs and NRs, could be controllably synthesized under visible light illumination. Owing to the advantages of microwave radiation, including a fast reaction rate, fast volume heating, good selectivity, simplicity, and a short reaction time, it has already been widely employed to produce nanomaterials of various shapes. In 2011, Ji’s research group confirmed that spherical particle-decorated single-crystalline Te NTs in the shuttle shape could be rapidly built by a microwave-assisted method in the presence of polyols as the reactive medium [13]. Similarly, by using tellurium diethyldithiocarbamate as the Te source with polyethylene glycol and water as the solvents, Te NTs were obtained through a microwave reflux method by Guan et al [2].

In one word, Te NTs synthesized by liquid-phase methods have their characteristics and advantages, as well as limitations and possibilities for further improvements. It is noteworthy that the shape and size of Te NTs can be customized by changing the solvent, reducing reagent, temperature, pH, additives, assistant treatment, and others, thus allowing them to work well in specific applications. In addition, it must be pointed out that the customizing of Te NTs is not simply dominated by any of the above parameters, but a synergy of them.

#### 2.1.2. Gas-Phase Methods

Compared with the widely used liquid-phase methods, the gas-phase synthesis of Te NTs also shows its uniqueness. Given that no template or other compound was introduced in the growth step of gas-phase methods except for the Te-containing precursors, the thus-obtained Te NTs were of extremely high purity. The gas-phase methods mainly include physical vapor deposition (PVD) with a sublimation–condensation process, and Figure 3a shows a schematic diagram of a typical experimental facility that produces Te NTs using PVD. Therefore, they heavily relied on the evaporation/deposition temperature, pressure, reaction time, carrier gas flow rate, and the used substrate.

In 2004, C. Me’traux et al. performed two distinct PVD processes to tailor the nanostructures of Te in the argon-atmospheric furnace in a magnetic field and under vacuum, and thus Te NTs with a controllable aspect ratio and Te blades/NRs were observed, respectively. As shown in Figure 3, by employing PVD in the absence of a magnetic field, hexagonal column-shaped Te NTs were obtained by Sen et al. by optimizing the source/deposition temperatures (550 °C/<200 °C) and argon flow rate. It was found that higher deposition temperatures within 350–430 °C could produce Te microrods and Te NWs [31,32]. Li et al. have demonstrated that under the argon atmosphere and high temperature (~560 °C for 2 h), powdery Te could be rapidly vaporized and transported before being deposited in the cooler region [33]. Finally, single-crystalline prismatic Te NTs were obtained after cooling the quartz tube to the room temperature, of which the size and geometry were found to be varied in the dosage of Te powder, temperature, and argon flow rate.

Among all parameters involved in PVD methods, the selection of the substrate has drawn increasing attention in the nearest decade, especially considering the scalable production of well-controlled Te NTs. On a Si (100) substrate, Kim et al., for the first time, reported highly pure single-crystalline Te NTs with triangular cross-sections by applying a thermal evaporation process [10]. After evaporating the Te powder at 350 °C and postcondensing at 150–200 °C, the Te NTs were observed downstream of the argon flow, while replacing the substrate with Si (111) or sapphire (0001) formed Te NWs and NRs instead. Te NTs could also be synthesized on an alumina substrate; however, pure Te NTs were formed within a certain temperature and time only. Besides the nonmetal substrates, substrates based on metal were also reported. For example, Kumar et al. reported that Te NTs could be grown on the Ag/Au nanoparticle (NP)-decorated Si substrates via a 100 °C vacuum deposition technique [1]. Their findings indicated that the introduction of metallic NPs under high vacuum (~10^−5^ Torr) increased these NTs’ density with the reduced diameter compared with those prepared at atmospheric pressure.

Even though gas-phase methods usually have a higher synthesis simplicity and environmental benignancy than the liquid-phase methods, the further development of the former for large-scale production is still limited by the size of furnaces and the cost-inefficiency of high temperature/vacuum treatments.

### 2.2. The Growth Mechanisms of Te NTs

Production of Te NTs with a desired size and shape under controllable conditions requires a fundamental understanding of their growth mechanisms. To date, the most widely recognized growth mechanisms for the formation of nanostructured Te are Ostwald ripening and oriented attachment [34,35]. Suppressed by the Gibbs adsorption and interfacial complexions, Ostwald ripening induced the formation of core/shell nanoprecipitated SnAg_0.05_Te-x%CdSe to display decreased thermal conductivity as their phonon scattering was retarded while maintaining a high carrier mobility [35]. Oriented attachment was performed to design PbTe nanocrystals with one-dimensional linear and zigzag and 2D square/honeycomb superstructures, which was passivated by their high surface reactivity to enable their flat bands and the Dirac cones in the valence and conduction bands to be moderately optimized. Inspired by these two, the specific growth mechanisms of Te NTs are well developed and significantly varied in preparation methods, including the seed-induced growth (SIG)/nucleation–dissolution–recrystallization growth (NDRG), and the helical belt template (HBT)/scrolling growth (SG). Details about these mechanisms are discussed in this section. Figure 4 shows the growth process of Te NTs.

#### 2.2.1. The SIG/NDRG Mechanism

As reported in the previous literature, a cylindrical seed was observed inside all Te NTs, as shown in Figure 5, leading Mayers and Xia to propose that Te NTs were possibly formed through concentration depletion at the surfaces of seeds, which was later categorized as the SIG mechanism, as follows [7]. At the first stage of the reaction, a large number of Te seeds were rapidly formed thanks to the synergy of a homogeneous nucleation process and the redox reaction between orthotelluric acid and ethylene glycol. Subsequently, the circumferential edges of each seed that had relatively higher free energy than other sites on the surface would preferentially grow larger with the newly-generated Te atoms in the system. As soon as the seed started to grow, the mass transport of Te atoms would also develop until undersaturation or complete depletion centered the (001) planes of each growing seed, to eventually form hollow Te NTs. Except for the polyol reflux method, the SIG mechanism also worked for the growth of Te NTs by the hydrothermal method and thermal evaporation [10,36]. Particularly, it was found that the hotter/cooler regions in the reaction furnace tended to induce the undersaturation/supersaturation of Te atoms, correspondingly favoring the formation of Te NTs/NWs.

In 2005, the hydrothermal synthesis of Te NTs based on the formamide reduction of Na_2_TeO_4_·2H_2_O at 160 °C was reported by Xi et al. (Figure 6) [37] Different from the inside of cylindrical seeds and Te NBs, some sphere-like trigonal Te crystals were observed through the evolution of Te NTs from NPs, categorized as the NDRG mechanism. Therefore, they believed that, initially, lots of Te NPs were formed and gradually dissolved to some free Te atoms (Figure 6a), which could act as the nucleation seeds, and were renewably transferred onto the surfaces of other undissolved sphere-like NPs (Figure 6b); then, these nanoclusters would slowly grow to form the groove-like NRs due to their structural anisotropy and undersaturation in the central growing regions (Figure 6c). The thus-formed NRs were apt for growing faster on the tuber axis direction than their circumferential direction before reaching a balance, thus eventually leading to the formation of NTs (Figure 6f). This NDRG mechanism was also confirmed by studies using the solvothermal method and thermal evaporation [22,31].

#### 2.2.2. The HBT/SG Mechanism

As displayed in Figure 7, single-crystalline Te NTs could be hydrothermally synthesized via the in situ disproportionation of Na_2_TeO_3_ at 180 °C within the pH range of 12–12.5 (ammonia); however, the following TEM observations indicated that no seed was found at the ends of any Te NTs. Interestingly, some helical nanobelt-lined Te NTs and tail-like helical belt-ended Te NTs were observed in other TEM images, revealing the HBT mechanism. In this case, it was possible that the early-formed TE NBs served as the template and continued rolling up around instead of epitaxially growing along its edges and joining both the side edges of itself, to finally form NTs [38].

He et al. [12] demonstrated that Te NTs with a shuttle-like morphology and sharp tips were formed based on a different mechanism considering the presence of trigonal TE crystal-possessed layer structures and TE NTs with scrolling edges, which were characterized by XRD, SEM, and TEM, through the shape evolution process (Figure 8). They clearly illustrated the amorphous TE NPs were yielded at first and gave birth to other easily aggregated nanostructured Te on their surface thanks to the functionalities of peripheral serine. Next, compared with other side groups of adopted amino acids, alcohol groups of serine could induce higher surface tension to grow thinner nanofilms in situ, which turned to curl and grow along with the Ostwald ripening process, resulting in NTs with a smoother surface. Due to the interaction of the alcohol groups and the carboxyl ones of two adjacent amino acids, more amorphous NPs were adsorbed on NTs and led to epitaxial layer growth, finally forming the shuttle-like Te NTs on the basis of the SG mechanism. To synthesize Te NTs with better performance, many efforts have been devoted to the deep learning networks of the SG mechanism, such as that by Zhang et al. [24], who witnessed similar growth steps during the solvothermal transformation of powdery Te to Te NTs, and in Ji’s work [13], which indicated that this mechanism matched the microwave-assisted polyol synthesis of shuttle-shaped single-crystalline Te NTs well.

It was shown experimentally that sometimes the formation of Te NTs did not completely follow one of the above four mechanisms, or rather a combination of multiple mechanisms. For example, Guan et al. [2] synthesized Te NTs by microwave refluxing diethyldithiocarbamate tellurium in the mixed polyethylene glycol/water, whose results suggested that the growth of Te NTs could be clearly explained by both the SIG mechanism and the NDRG mechanism. Moreover, there must be other mechanisms regarding the formation of Te NTs, so more comprehensive studies are needed to provide new insights.

## 3. Property Control

Te is well known as a p-type narrow bandgap (0.35 eV, direct) semiconductor that lacks centrosymmetry; thus, the electrical [39], optical [13,39,40], magnetic [23,33], and other properties [1,41] of its NTs are greatly controlled by their geometric, structural, physical, and chemical features. Rational optimization of these factors is important to ensure the effective uses of Te NTs in electronic and optoelectronic applications.

### 3.1. Electrical Properties

Given that Te is a metalloid with relatively large spin–orbit coupling [42], Te NTs are endowed with the highest electrical conductivity among inorganic elements (2 × 10^2^ S·m^−1^), p-type narrow bandgap, and high structural rigidity over flexible NBs and NWs, and they generally have highly stable electrical properties depending on their size and heteroatom doping. For another ultralong submicron Te NT, it was found that the trace level doping of Na could thermally scatter their weakly-bonded lattice with ease, thus enabling the resistivity of these Te NTs to decrease upon cooling (5–300 K). By employing an individual of them as the building block, a nanodevice was built through focused-ion-beam deposition to exhibit a quadratic temperature-dependent resistivity, of which the room-temperature resistivity and the ratio of 5 K resistivity/room-temperature resistivity could reach 9.854 μΩ and 0.47, respectively [39]. Te NTs with an average grain size <10 nm and wall thickness range of 15–30 nm were embedded into a field effect transistor, whose mobility was decreased to ~0.01 cm^2^/V·s, as its phonon scattering was dominated by the Te NT lattice that could decrease the thermal conductivity for the increased thermoelectric figure of merit [23]. Additionally, its field effect mobility was temperature-dependent and obeyed the Conwell–Weisskopf relationship within the temperature of <250 K. For another Te NTs with an easily tunable diameter (40–100 nm), by using the solvothermal method, their surface-to-volume ratio and crystallinity were optimized to fill their surface trap states and crystalline defects with more photo-generated holes. As a consequence, a high photoresponsivity of 1.65 × 10^4^ A·W^−1^ and photoconductivity gain of 5.0 × 10^6^% were observed on the optoelectronic nanodevice based on these Te NTs [41].

### 3.2. Optical Properties

As most Te NTs have a single-crystalline structure, their optical properties can be modified through geometric control and chemical treatments. For example, Yu’s research group [26] reported the single-crystalline trigonal Ne NTs with sloping and hexagonal cross-sections grew along the (001) direction and had the outer diameters/wall thicknesses/lengths within 100–500 nm/50–100 nm/150–200 μm. Further, the 365 nm photoluminescence excited these Ne NTs to present blue–violet emissions (390–550 nm) for the first time, which was highly related to the thickness of the nanostructures and crystallization behavior of the solvothermally reduced Te NTs [40]. Replacing the distilled water with absolute ethanol, hydrothermally prepared Te NTs which had sloping cross-sections, open ends, and relatively short lengths of 30–50 μm were also well formed. These Te NTs were found to present a concentration-dependent excitation/emission, attributed to their thickness and highly anisotropic crystallization. It was also reported that the further decoration of spherical Te NPs on the shuttle-shaped Te NTs could increase their chirality, considering the inherently helical chain structure with two ends, to induce a brand new strong red emission, beneficial for nano-optical applications [13]. Except for geometric modifications, the oxygen-related defects formed on the hexagonal column-shaped Te NTs favored the electron radiation transition from the p-antibonding triple of conduction band to the p-bonding triple of valence band in the latter, helping them to obtain a broad photoluminescence peak at ~532 nm [43].

### 3.3. Magnetoresistance Properties

Ever since the positive magnetoresistance effect at low temperature was recorded on the Te microtubes by Li et al. in 2003 [33], only a few attempts have been made to explore the magnetoresistance properties of Te NTs. Later, Rheem et al. [23] demonstrated that the unique magnetoresistance properties behavior could be observed on the Te NTs fabricated by galvanic displacement, which presented a magnetoresistance ratio up to 37% (260 K); however, the related mechanism was unclear. The latest analogues of Te NTs including the layered transition metal dichalcogenides NiTe_2_, PdTe_2_ and PtTe_2_, irrespective of which did or did not host Weyl or Type-II Dirac fermions, had high intrinsic carrier mobility, but its high purity was the prerequisite to observe maximal magnetoresistance effects [44]. In this case, the strong interconnection between carrier mobility and magnetoresistance contributed to the temperature dependence for an individual sample or the difference between the samples via forcing carriers on Landau orbits by the applied transverse field.

In addition to the above, Te NTs also have many characteristics, such as sensitive gas sensing [1,45], outstanding mechanical properties [6], and stability [7,26]. Note, doped heteroatoms, especially metals, could endow Te NTs with additional excellent properties, such as high thermopower, small thermal conductivity [16,46,47], antibacterial ability [48], roll-to-roll processability [49], and catalytic ability [46,50], etc.

## 4. Applications

Benefiting from the above characteristics, Te NTs are proven to be versatile and applicable in sensing and decontamination [1,2,45,51], energy storage [52,53], thermoelectrics [16,47,54,55,56], and templating for catalysts [46,50]. Particularly, Te NTs possess certain benefits for these applications that include a high specific surface area, tailorable charge transfer/transport, and the ability to heterostructure with other nanomaterials [18,49,50].

### 4.1. Sensing and Decontamination

Recently, various sensors based on semiconductors have been extensively studied due to their small size, relative simplicity, and low cost. As a p-type semiconductor, elemental Te can respond to the ppb (nM) level of many gassy compounds (ions) at room temperature, and thus Te NTs are expected to offer much a lower limit of detection (LOD) because of the high specific surface area resulted from their hollow tubular structure. In principle, Te NTs can serve as the resistors in gas sensors to present resistivity change in response to the chemical environment change upon absorbing specific atoms in gases. To the best of our knowledge, Te NTs were reported to be sensitive to both oxidative (NO, NO_2_, and CO) and reductive (NH_3_ and H_2_S) gases, for which the further doping of Au, Pt, and Ag could obtain even higher anti-interference. On exposure to oxidizing NO (reducing NH_3_ or H_2_S), the resistance of a vacuum-deposited Te NT-built gas sensor [1] decreased (increased), which was ascribed to the surficial-adsorbed O-induced TeO_2_ formation from Te and electrons trapping that could increase hole density, and conductivity was facilitated (passivated) by NO (NH_3_ and H_2_S). Affected by the similar redox reaction, the hexagonal column-shaped Te NTs [45] enabled the thus-constructed gas sensor to detect CO (30 ppm) and NO_2_ (3 ppm) within 11 min and 6.5 min, respectively, at room temperature. Based on another microwave-assisted polyol reflux-synthesized cylindrical Te NTs [2], a chemiresistive sensor exhibited an ultralow LOD of NO_2_ (≥0.5 ppb within ≥3 min), even in the real environment, as well as good selectivity toward CH_4_, NH_3_, CO, and H_2_ due to the much higher electron affinity of NO_2_ when adsorbed on this sensor(Figure 9). Except for gases, Te NTs are also viable for sensing ions through the redox reaction. For example, Wei et al. [51] developed a trigonal Te NTs-incorporated agarose gel membrane that had a high surface area and strong hybridization between the p (Te) and d (Hg) electronic states at the valence band edge (*H*_f_ (HgTe) = 244 kJ·mol^−1^) to act as an optical sensor for Hg^2+^. Even for Hg^2+^ spiked in real environment water, a low LOD down to 10 nM and a high removal above 97% could be selectively and quickly obtained through the following displacement reaction.
2Hg^2+^ + Te + 3H_2_O → 2Hg +TeO_3_^2−^+ 6H^+^

In addition to gases and ions, Te NTs could also be employed to fabricate working electrodes for photodetection. As shown in Figure 10, the epitaxial growth of Se on Te NTs formed the Te@Se roll-to-roll NTs [49], in which the increased optical absorption, enhanced built-in electric field, and suppressed carrier transport contributed to the sharp increases in the photocurrent density (7.79 μA·cm^−2^) and photoresponsivity (98.8 μA·W^−1^) and excellent tolerance to aqueous solutions (HCl, NaCl, and KOH) of the photodetector. Zhang et al. [18] recently reported the Bi quantum-dots-decorated Te NTs (Te@Bi) van der Waals heterojunctions that exhibited a high photocurrent density (16.87 μA·cm^−2^) and photoresponsivity (142.97 μA·W^−1^), which was proven to result from the interfacial plasma effects and the van der Waals between Te NTs and Bi quantum dots.

### 4.2. Energy Storage

With respect to energy conversion and storage, Te seems one of the most promising electrode candidates due to its high electrical conductivity and theoretical volumetric capacity (2621 mA·h·cm^−3^). Additionally, compared with the same main group elements (such as sulfur and selenium), Te exhibits relatively smaller volume expansion and faster charge–discharge kinetics, showing particular potential as an attractive electrode material for Li-ion batteries. Therefore, it is expected that the high specific surface area, single-crystallinity, flexibility, and film-forming ability will increase the electrochemical active sites in Te NTs to present faster ion diffusion and better charge storage. In the flexible all-solid-state Li-ion batteries constructed with a novel Te NTs-grown carbon fiber cloth cathode [52], the embedding of the hexagonal phase Te NTs with high single crystalline quality and solvent-free feature increased the electrochemical–structural stability, electrical conductivity, and flexibility. Thus, the high gravimetric capacity of 316 mA·h·g^−1^ and volumetric capacity of 1979 mA·h·cm^−3^ (@100 mA·g^−1^) after 500 cycles were obtained. As shown in Figure 11, after being uniformly deposited on a nanofibrillated cellulose film as an anode for Li-ion batteries, Te NTs [53] favored the electrode–electrolyte contact, ion diffusion, structural integrity maintenance, and consequently reached a high volumetric capacity (1512 mA·h·cm^−3^ @200 mA·g^−1^), high capacity retention (104% over 300 cycles), and excellent rate performance (833 mA·h·cm^−3^ @100 mA·g^−1^).

### 4.3. Thermoelectrics

Thermoelectrics are a kind of electronics that can enable heat to be converted into electrical current, which is considered one of the solutions to the severe environmental issues arising from the increasing overuse of energy. As is well known, the crucial issue in thermoelectrics’ study is to increase *Z*·*T* (= *α*^2^*T*/*ρk*), where *Z*, *T*, *α*, *ρ*, and *k* represent the figure of merit (K^−1^), the absolute temperature (K), the Seebeck coefficient (V·K^−1^), electrical conductivity (S·m^−1^), and thermal conductivity (W·m·K^−1^), respectively [54]. In other words, simultaneously achieving a high power factor (*α*^2^/*ρ*) and low thermal conductivity is the prerequisite for high-performance thermoelectrics [16]. Jung et al. [55] proposed the Sb-doped Te-rich NTs with hollow hexagonal cross-sections that could act as the phase-change memory switch for an electrical pulse. Benefiting from their small cross-sectional area, geometrical effect on the heat transport, high electrical conductivity, and low thermal conductivity, a much lower writing current was needed for the induction of amorphization voltages pulses. As they are still evolving, Te NTs will be frequently utilized as the secondary components for fabricating thermoelectrics in the 15 years ahead. It was reported by Kyung et al. [47] that by growing Bi_2_Te_3_ on Te NTs through polyol refluxing, many single-crystalline Bi_2_Te_3_ NPs were bonded on the nanotubular structure that reduced the lattice thermal conductivity of the thus-generated powdery nanocomposite due to the increased phonon scattering. Furthermore, the use of Te NTs as the substrate to grow Bi_2_Te_3_ realized the formation of a core/shell heterostructured Te/Bi_2_Te_3_ composite, of which the rough serrated interfaces and hollow structures synergistically enhanced the phonon scattering inside it to present as a low thermal conductivity of 0.43 W·m·K^−1^ within 300–400 K [56] (Figure 12).

### 4.4. Templating for Catalysts

In recent years, there has been a trend toward nanoarchitectured catalysts enabled by the sacrificial templating of Te NTs, which can act as a physical scaffold and reactant to induce the resultant to form in the same geometry as Te NTs. For inorganic materials that are unlikely to assemble into tubular nanostructures by using existing technologies, the use of Te NTs as the sacrificial template turns out to be a reliable route to confer them with hollow interiors and nanotubular geometry, as well as a highly tailorable shape and size. Making use of uniform dispersion, geometric stability, and temperature-dependent reactivity, Te-NTs were employed as a self-sacrificial template to generate various shish-kebab-like CdS-Te@(Pt, Pd) multiheterostructures [50] (Figure 13). The increased and stable photocatalytic production of H_2_ of them was attributed to the intimate interaction between CdS-Te and other components, along with more active sites, improved separation of photogenerated carriers, and enhanced light adsorption. Te NTs are also desirable templates of electrocatalysts because of their well-controlled size and morphology to ensure as ideal a geometry as possible, thus leading to the optimized electrocatalytic activity. Lou et al. had devised a facile route to produce porous Pt NTs from the bamboo-like Te NTs template, and the unique tubiform structure and nanoporous framework helped the former to exhibit excellent electrocatalytic activities toward methanol oxidation, alternatively available to direct methanol fuel cells [46].

## 5. Current Challenges and Outlook

Te NTs are single crystalline 1D hollow semiconductor nanomaterials that could change the landscape in many fields of science and technology. Although the development is rapid, it is nevertheless in graduated stages and some issues remain unsolved. Firstly, it is a prerequisite to have an unambiguous and generally acceptable definition of those shapes of Te NTs, including cylindrical, prismatic, bamboo-like, shuttle-like, or otherwise. Only by making the distinction clearer can the literature and the applications of Te NTs be made easier to understand, repeat, and optimize. Secondly, regardless of the synthetic strategies, relatively broader distributions of length, diameter, wall thickness, and aspect ratio are yielded for most Te NTs, correspondingly covering a very wide range of electrical, optical, magnetoresistance, and other properties. Therefore, a promising application for Te NTs based on one certain synthesis may also be feasible for that synthesized by another one. In other words, it is highly required to find out these fabrication method counterparts so that the engineering of Te NTs to present well-defined characteristics is more experimentally accessible to researchers in the community, as well as more effective applications that can be achieved more easily.

For sensing and decontamination, energy storage, thermoelectrics, and templating for catalysts, the disadvantage of Te NTs is the lack of economic scale-up fabrication. Unlike carbon NTs, currently, the gram-scale production of Te NTs can hardly satisfy the demands of continuously developing fundamental research and practical applications. Therefore, more efforts need to be paid to the development of the fabrication methods, especially the relatively low-cost and efficient routes, including polyol reflux, the electrochemical method, and thermal evaporation. On the other hand, although it is difficult to produce as many Te NTs as their 1D nanomaterials, the uniqueness of the former can still guarantee their application in specific fields. In addition, currently, the reportage of Te NTs in biomedicine is rare, although it may only require a small amount, far below the clinical poisoning threshold, of Te NTs as the additives.

The electrical, optical, magnetoresistance, and other properties of Te NTs are highly related to their size, shape, and chemical defects. Experimentally isolating their influence is challenging, as the reports about precisely controlling the geometry of Te NTs are rare, and so deeper theoretical studies are needed to reveal more insights. Currently, knowledge about the interplay between these factors is still limited, which strictly hinders the comprehensive improvement of Te NTs in various applications, let alone the finding of new properties in them for new applications. In addition, the growth mechanisms of Te NTs have been frequently reported; however, the exclusivity/compatibility among some of them is rarely studied. Moreover, the detailed investigation and understanding of growing heterostructured NTs of Te or the alloying Te NTs with other elements/compounds are highly anticipated.

The current scope of Te NTs applications has already covered a wide range of topics, and it is still expanding as coronavirus disease 2019 continues to spread worldwide. Just like other 1D Te nanomaterials, Te NTs are expected to engineer and improve the properties of more and more nanomaterials. Because of their good uniformity and tailorable geometry, Te NTs may serve as a new class of sacrificial templates for many applications. Overall, the study of Te NTs is very young, but still has great potential if current challenges can be tackled and barriers can be removed. Thus, the desirable merits of Te NTs will be understood in a deeper way and more efficiently utilized.

## Figures and Tables

**Figure 1 nanomaterials-12-02151-f001:**
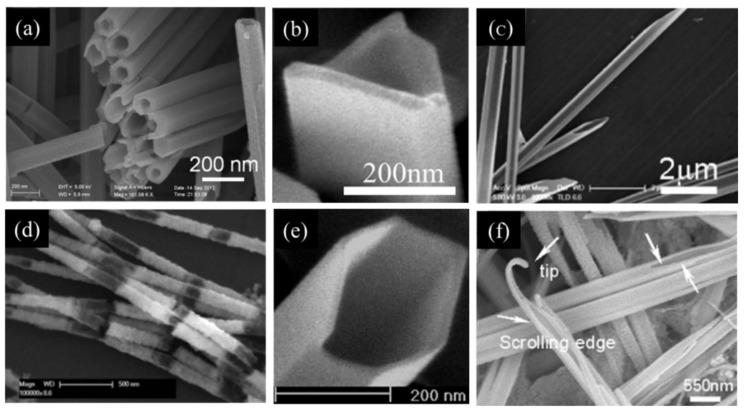
SEM images of Te NTs with different shapes. (**a**) Cylindrical Te NTs. Reprinted with permission from [8]. Copyright 2016, Wiley-VCH Verlag GmbH & Co. KGaA, Weinheim, Germany. (**b**) Individual Te NTs with triangular cross section. Reprinted with permission from [10]. Copyright 2006, American Chemical Society, Washington, DC, United States. (**c**) Te NTs synthesized with sloping cross-sections. Reprinted with permission from [26]. Copyright 2008, American Chemical Society, Washington, DC, United States. (**d**) Bamboo-like Te NTs. Reprinted with permission from [11]. Copyright 2014, Wiley-VCH Verlag GmbH & Co. KGaA, Weinheim, Germany. (**e**) Te NTs with hexagonal cross-section. Reprinted with permission from [10]. Copyright 2006, American Chemical Society, Washington, DC, United States. (**f**) Shuttle-like Te NTs. Reprinted with permission from [12]. Copyright 2005, American Chemical Society, Washington, DC, United States.

**Figure 2 nanomaterials-12-02151-f002:**
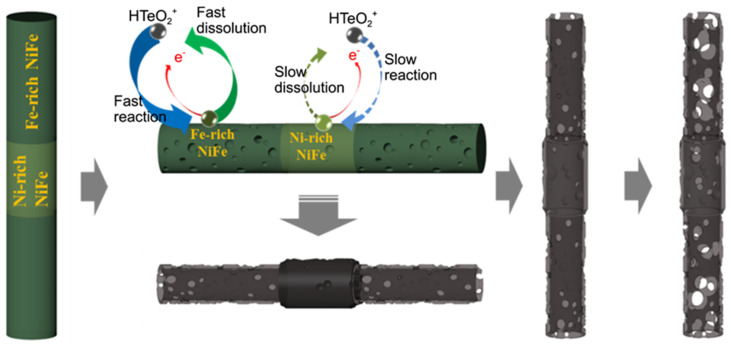
Schematic diagram of the bamboo-shaped Te NTs produced by the galvanic displacement reaction of a segmental NiFe nanowire with Ni-rich and Fe-rich NiFe segments in an electrolyte consisting of HTeO^2+^ ions in acidic HNO_3_ solution. Reprinted with permission from [11]. Copyright 2014, Wiley-VCH Verlag GmbH & Co. KGaA, Weinheim, Germany.

**Figure 3 nanomaterials-12-02151-f003:**
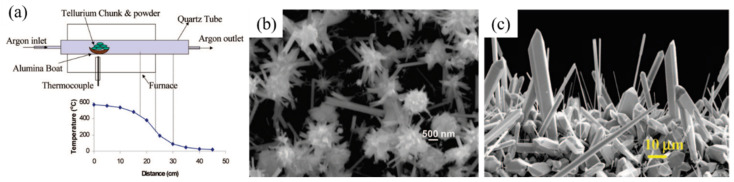
(**a**) Schematic diagram of experimental device for preparation of Te nanotubes by physical vapor deposition shows gas flow arrangement in tubular furnace. Different temperature zone in the furnace along the flow of the gas is also shown. (**b**) SEM of Te nanotubes growing radially from Te sphere in the temperature zone of 200 °C. (**c**) SEM image of Te microrods grown in the temperature zone of 400–350 °C. Reprinted with permission from [31]. Copyright 2008, American Chemical Society, Washington, DC, United States.

**Figure 4 nanomaterials-12-02151-f004:**
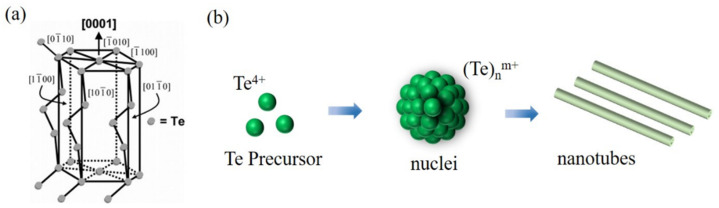
(**a**) Schematic representation showing the hexagonal crystal structure of tellurium. Reprinted with permission from [10]. Copyright 2006, American Chemical Society, Washington, DC, United States. (**b**) A schematic illustration of the reaction pathways that lead to Te nanotubes.

**Figure 5 nanomaterials-12-02151-f005:**
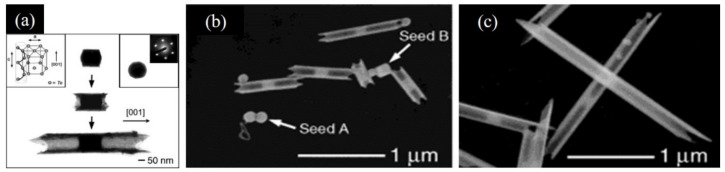
Te NTs that were synthesized by refluxing a solution of orthotelluric acid in ethylene glycol. (**a**) TEM images of tellurium nanotubes at three different stages of growth. The inset at the upper left corner shows an illustration of the crystal structure of trigonal tellurium that contains helical chains of tellurium atoms packed parallel to each other along the c-axis. The inset at the upper right corner gives the cross-sectional TEM image of a cylindrical seed, together with a microdiffraction pattern obtained by focusing the electron beam on this seed along the [001] direction. (**b**,**c**) SEM images of tellurium nanotubes at different reaction times. (**b**) 4 min, (**c**) 6 min, respectively. Several seeds, as indicated by arrows in (**b**), were still present in the early stage of this process. These seeds had two spatial orientations, with their c-axis perpendicular (type A) or parallel (type B) to the solid support. The two type seeds aggregated into a doublet during the evaporation of solvent. Reprinted with permission from [7]. Copyright ©2007 WILEY-VCH Verlag GmbH & Co. KGaA, Weinheim.

**Figure 6 nanomaterials-12-02151-f006:**
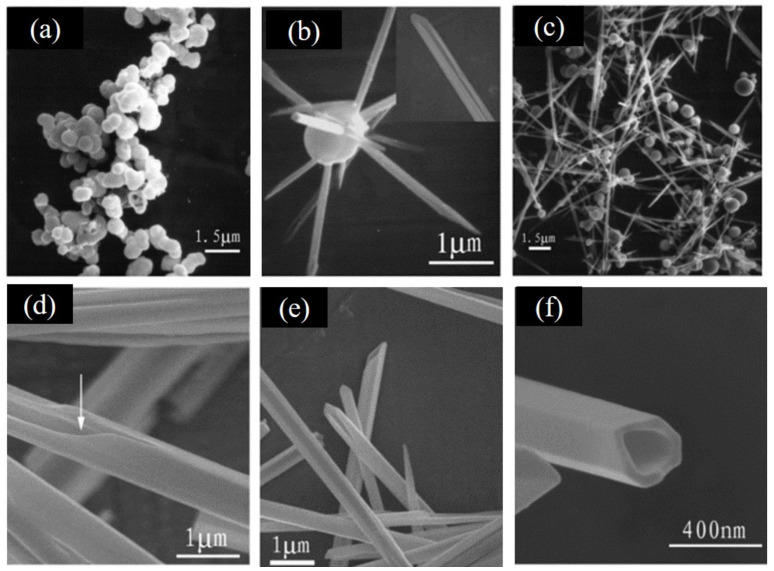
SEM images of five Te NTs samples collected after hydrothermal treating for (**a**) 5 h, (**b**) 8 h, (**c**,**d**) 12 h, (**e**) 16 h, and (**f**) 20 h, supporting the NDRG mechanism. Reprinted with permission from [37]. Copyright 2004, American Chemical Society, Washington, DC, United States.

**Figure 7 nanomaterials-12-02151-f007:**
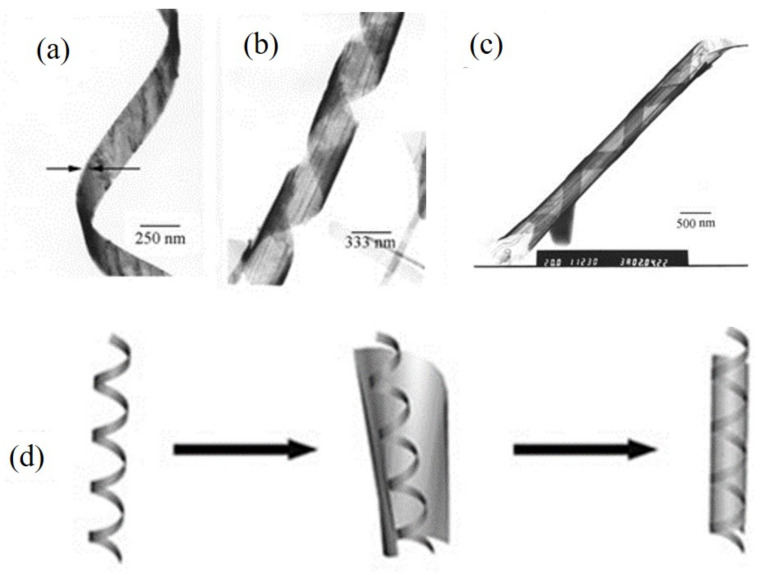
(**a**–**c**) Representative TEM images of the as-synthesized Te NBs and NTs, showing their geometric shapes. (**a**) Twisted Te nanobelt. (**b**) Helical nanobelt, displaying the helical twisting property of the NBs. (**c**) A typical belt-rolled Te nanotube. (**d**) Schematic illustration of the proposed HBT mechanism. Reprinted with permission from [38]. Copyright 2002, WILEY-VCH Verlag GmbH & Co. KGaA, Weinheim.

**Figure 8 nanomaterials-12-02151-f008:**
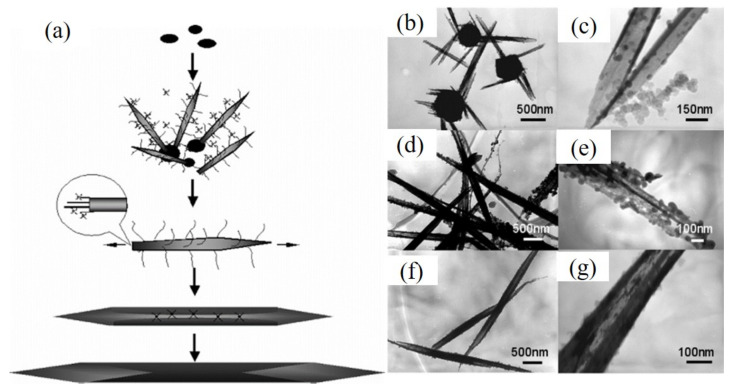
(**a**) The SG mechanism for the formation of shuttle-like Te NTs using serine as additive. (**b**–**g**) TEM images of Te NTs obtained at different stages: (**b**) and (**c**) 2 h, (**d**) and (**e**) 4 h, (**f**) and (**g**) 12 h. Reprinted with permission from [12]. Copyright 2005, American Chemical Society, Washington, DC, United States.

**Figure 9 nanomaterials-12-02151-f009:**
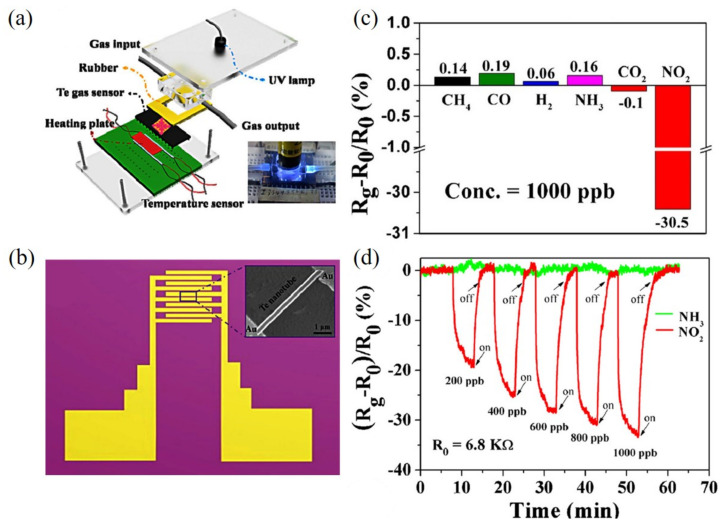
(**a**) The schematic and photographs (inset) of a customized chemiresistive sensor based on Te NTs, and (**b**) the schematic of an independent interdigitated electrode. Inset in (**b**) is a SEM picture of one Te nanotube on an interdigitated electrode. (**c**) Real-time response of the gas sensor toward NH_3_ and NO_2_, and (**d**) response of the sensor to different gases that have a concentration of 1000 ppb. Reprinted with permission from [2]. Copyright 2014, Elsevier B.V., Amsterdam, the Netherlands.

**Figure 10 nanomaterials-12-02151-f010:**
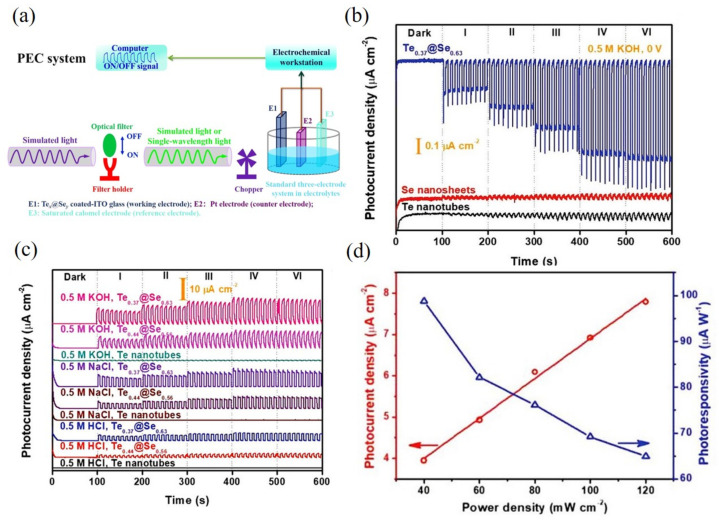
(**a**) A typical photoelectrochemical system built for evaluating the photoresponse behavior of the Te_x_@Se_y_-based photodetector in electrolytes. (**b**) Self-powered photoresponse behaviors of Te NTs, Se nanosheets, and Te_0.37_@S_e0.63_ in 0.5 M KOH under simulated light. (**c**) Size effect of Se on Te NTs on the photoresponse performance of Te_x_@Se_y_-based photodetector under simulated light in 0.5 m different electrolytes. (**d**) Fitting curve of *P_ph_* and calculated photoresponsivity (*R_p__h_*) of Te_0.37_@S_e0.63_ under simulated light with various power densities in 0.5 M KOH. Reprinted with permission from [49]. Copyright 2007, WILEY-VCH Verlag GmbH & Co. KGaA, Weinheim.

**Figure 11 nanomaterials-12-02151-f011:**
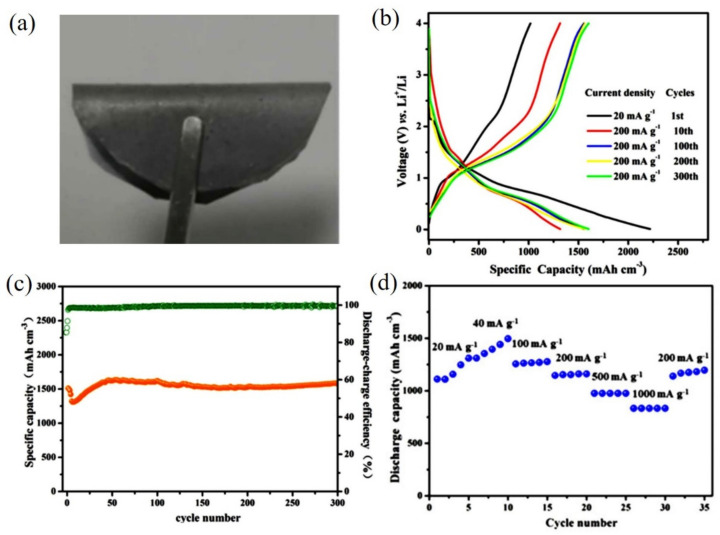
(**a**) The photos of the flexible Te NTs-based electrode under bent state; (**b**) Representative charge–discharge curves of the 1st, 10th, 100th, 200th, and 300th cycles; (**c**) Cycle performance and Coulombic efficiency at 200 mA g^−1^ for 300 cycles, and (**d**) Rate performance in the rate range of 20–1000 mA g^−1^. Reprinted with permission from [53]. Copyright 2021, Licensee MDPI, Basel, Switzerland.

**Figure 12 nanomaterials-12-02151-f012:**
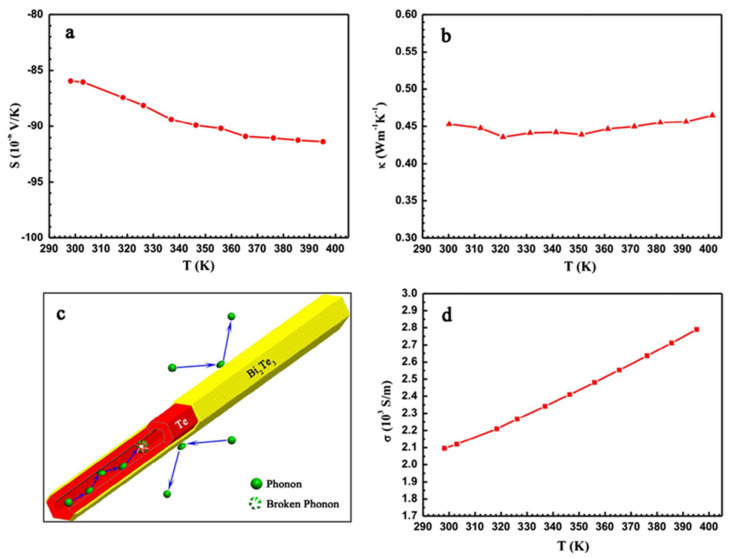
(**a**) Seebeck coefficient and (**b**) Thermal conductivities of the Te/Bi_2_Te_3_ core/shell heterostructure NTs depending on the temperature; (**c**) Mechanism of phonon scattering in the center or on the surface of the NTs; (**d**) Electrical conductivities based on temperature. Reprinted with permission from [56]. Copyright 2014, Elsevier Ltd., Amsterdam, The Netherlands.

**Figure 13 nanomaterials-12-02151-f013:**
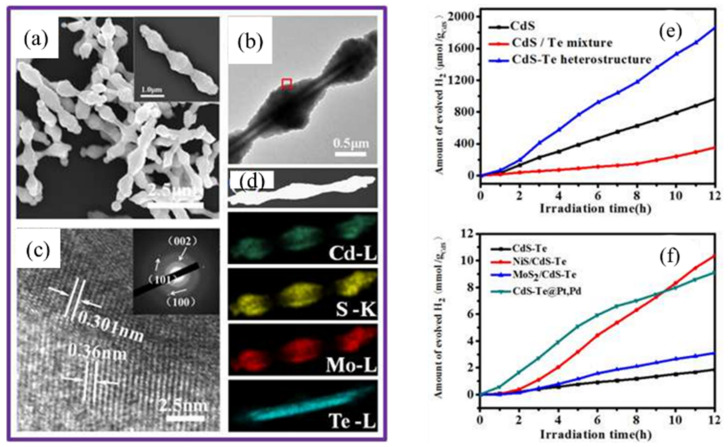
(**a**–**d**) MoS_2_/CdS-Te sample characterized with (**a**) SEM, (**b**) TEM, (**c**) HRTEM of the area marked with a square in (**b**) (the inset showing the corresponding SAED pattern), and (**d**) STEMEDS elemental mapping, showing the Cd (celadon), S (yellow), Mo (red), and Te (wathet). (**e**,**f**) Comparison of the photocatalytic activities of (**e**) CdS, CdS/Te mixture, and CdS-Te heterostructure, respectively; and (**f**) CdS-Te, NiS/CdS-Te, MoS_2_/CdS-Te and CdS-Te@Pt, Pd, respectively, in a lactic acid aqueous solution under visible light illumination. Reprinted with permission from [50]. Copyright 2015, American Chemical Society, Washington, DC, United States.

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
