# Peer review of "Tellurium Nanotubes and Chemical Analogues from Preparation to Applications: A Minor Review"

_nanomaterials, 2022, doi:10.3390/nano12132151_

Round 1

Reviewer 1 Report

The Manuscript covers contemporary achievements in the field of Te nanotubes and nanoscrolls synthesis and application. It is no doubt that the review would be of interest for broad readership of Nanomaterials journal. I would recommend it for publication after considering several comments:

1) The Authors claims that the description of morphology is still rather challenging (at the beginning of Section 5, p. 15). However, I think that the term ‘nanoscrolls’ has been already well-established (for example, please check the review with DOI: 10.3390/cryst10080654), and it is based on the layer scrolling mechanism. If the Authors implied ‘nanoscrolls’ by using ‘Analogues’ in the title, they should consider changing it.

2) To date, the most widely recognized growth mechanisms are Ostwald ripening and oriented attachment. Could the Authors relate the mentioned mechanism with these two? I’m not also complete satisfied with the order, in which mechanism are considered. It is obvious that SIG and NDRG mechanisms have common features, as well as HBT and SG do. 

3) Some figures (for example, fig. 4) contain ‘raw’ information (SEM images) that is not essential for mechanism illustration. The informative one should contain, for example, particle size distribution evolution with time, because it might give a key to contribution of the Ostwald ripening and the oriented attachment. The Authors should consider this possibility in case it has been done in the referenced paper.

4) It might be useful to consider thermal properties of Te NTs, and, what is more important, to compare it with other forms of Te existence (for example, with thin films or so called 2D Te). The question is: what is the point in making nanotubes or nanoscrolls?

5) Some minor comments are: (a) last paragraph of Subsection 3.3 and introduction to the Section 4 (on good mechanical properties, in particular) look speculative without appropriate references, please add them; (b) some minor misprints were spotted, see caption of fig. 1.

Author Response

The Manuscript covers contemporary achievements in the field of Te nanotubes and nanoscrolls synthesis and application. It is no doubt that the review would be of interest for broad readership of Nanomaterials journal. I would recommend it for publication after considering several comments:

1) The Authors claims that the description of morphology is still rather challenging (at the beginning of Section 5, p. 15). However, I think that the term ‘nanoscrolls’ has been already well-established (for example, please check the review with DOI: 10.3390/cryst10080654), and it is based on the layer scrolling mechanism. If the Authors implied ‘nanoscrolls’ by using ‘Analogues’ in the title, they should consider changing it.

Response: Sorry for our unclear statements. Honestly, ‘Analogues’ in the title was used to compare pure Te with other Te-containing composites such as Bi2Te3 NTs, CdTe NTs, Te@Bi van der Waals heterojunctions, and carbon-coated MoS1.5Te0.5 nanocables. We agree with the reviewer that the term ‘nanoscrolls’ has been already well-established, therefore we revised the title as “… Chemical Analogues…”, also most “scroll” were removed to avoid misleading (except for the scrolling growth).

Revision: Please kindly see the revisions in the revised manuscript.

2) To date, the most widely recognized growth mechanisms are Ostwald ripening and oriented attachment. Could the Authors relate the mentioned mechanism with these two? I’m not also complete satisfied with the order, in which mechanism are considered. It is obvious that SIG and NDRG mechanisms have common features, as well as HBT and SG do. 

Response: This comment is highly helpful. The mentioned two mechanisms are added and the already mentioned four mechanisms are revised, as advised.

Revision: “Production of Te NTs with a desired size and shape under controllable conditions requires a fundamental understanding of their growth mechanisms. To date, the most widely recognized growth mechanisms for the formation of nanosctructured Te are Ostwald ripening and oriented attachment [34, 35]. Suppressed by the Gibbs adsorption and interfacial complexions, Ostwald ripening induced the formation of core/shell nanoprecipitated SnAg0.05Te-x%CdSe to display decreased thermal conductivity as their phonon scattering was retarded while maintaining high carrier mobility [35]. Oriented attachment was performed to design PbTe nanocrystals with one-dimensional linear and zigzag and 2D square/honeycomb superstructures, which was passivated by their high surface reactivity to enable their flat bands and Dirac cones in the valence and conduction bands to be moderately optimized. Inspired by these two, the specific growth mechanisms of Te NTs are well developed and significantly varied in preparation methods, including the seed-induced growth (SIG)/ nucleation-dissolution-recrystallization growth (NDRG), and the helical belt template (HBT)/ scrolling growth (SG). Details about these mechanisms are discussed in this section.

3) Some figures (for example, fig. 4) contain ‘raw’ information (SEM images) that is not essential for mechanism illustration. The informative one should contain, for example, particle size distribution evolution with time, because it might give a key to contribution of the Ostwald ripening and the oriented attachment. The Authors should consider this possibility in case it has been done in the referenced paper.

Response: We agree with the reviewer that the “raw” information should not be presented regarding the mechanism illustration, so the related figures are deleted and re-organized to smooth reading. Besides, a schematic diagram of growing mechanism of Te NTs is added as shown below.

Revision: (The revised) Figure 4.

4) It might be useful to consider thermal properties of Te NTs, and, what is more important, to compare it with other forms of Te existence (for example, with thin films or so called 2D Te). The question is: what is the point in making nanotubes or nanoscrolls?

Response: We agree with the reviewer that thermal properties are important to other forms of Te. However, thermal properties of Te NTs that have been only occasionally reported, except their thermoelectricity and thermal conductivity.

As already reviewed in the Introduction, the point in making NTs is that “1D nanostructured Te has drawn tremendous attention due to its intriguing properties, such as excellent thermoelectricity, high piezoelectricity, fast photoconductivity, nonlinear optical effects, and high sensitivity for gas (such as NO, NO2, and CO, etc.) and ions [1-3]. The successful construction of 1D nanostructured Te counterpart to other 1D nanomaterials is strongly related to its size and shape. Size has an effect on its specific surface area and surface-to-volume atoms ratio, while shape affects not only the facet size but also the content of surface atoms [4-6]. In this case, 1D Te nanomaterials include Te NWs, Te NBs, and Te NRs that are of relatively simple shapes and are outperformed by Te NTs. Different from the other three, the existence of the inner surface of Te NTs can help them to stay clean and intact during the fabrication of Te NTs-based devices for enhanced performance. On the other hand, Te NTs of many shapes, such as cylinder, prismatic (hexagonal column, trigonal column, etc), bamboo-like, and shuttle-like, have been reported in recent years [7-12]. By tuning the Te-containing precursor, the surfactant, reaction temperature, reaction time, and reducing reagent, the aspect ratio, morphology, and architecture of Te NTs can be rationally designed [7, 13, 14]. To the best of our knowledge, the advantages of these Te NTs are varied in shapes, further integrating or chemically compositing which with other elements/compounds can facilitate their applications in specific areas. For example, Te NTs with different hollow structures can be engineered to generate some tubular and nanowire-in-nanotube advanced functional materials, Bi2Te3 NTs, CdTe NTs, Te@Bi van der Waals heterojunctions, and carbon-coated MoS1.5Te0.5 nanocables [15-18], have promising applications in gas/ion sensing, catalysis, photodetectors, and energy storage. These results are attributed to their high specific surface area, strong interpenetrating network, and good electron/ion transport.

Revision: As clearly explained above, there is no revision needed in this case.

5) Some minor comments are: (a) last paragraph of Subsection 3.3 and introduction to the Section 4 (on good mechanical properties, in particular) look speculative without appropriate references, please add them; (b) some were spotted, see caption of fig. 1.

Response: Sorry for our mistake, the relevant literature are added and the corresponding minor misprints are corrected as advised.

Revision: Last paragraph of Subsection 3.3, “In addition to the above, Te NTs also have many characteristics, such as sensitive gas sensing[1, 47], outstanding mechanical properties[6], and stability[7, 48]. Note, doped heteroatoms, especially metals, could endow Te NTs with more additional excellent properties, such as high thermopower, small thermal conductivity[16, 49, 50], antibacterial ability[51], roll-to-roll processability [52], and catalytic ability[49, 53], etc.

Section 4, “Benefiting from the above characteristics, Te NTs are proven to be versatilely applicable in sensing and decontamination [1, 2, 47, 54], energy storage [55, 56], thermoelectrics [16, 50, 57-59], and templating for catalysts [49, 53]. Particularly, Te NTs possess certain benefits for these applications that include high specific surface area, tailorable charge transfer/transport, and the ability to heterostructure with other nanomaterials [18, 52, 53].

Reviewer 2 Report

In this work, a minor review about the synthesis and nanostructure control of Tellurium (Te) nanotubes (NTs), and the recent progress of research into Te NTs is summarized.

In general, the work is well structured. On my opinion section 3 is the poorest section, since authors go directly to the applications, without discussing deeper the basis of the physical properties.

I include some more comments:

1) Common reader would appreciate that caption Figures will be referenced to get fastly to the original works.

2) Section 3 composed of electrical, optical and magnetoresistance is the poorest section. Authors go directly to describing specific works, without discussing the basis of their physical properties. Reader would appreciate to know them properly in this section or at least to know the best works in the state of art about this.

3) Section 3.1: What is the reference of this assertion?

By employing an individual of them as the building block, a nanodevice was built through focused-ion-beam deposition to exhibit a quadratic temperature-dependent resistivity within 5-300 K, of which the room-temperature resis-tivity and the ratio of 5 K resistivity/room-temperature resistivity could reach 9.854 μΩ and 0.47, respectively.

Author Response

In this work, a minor review about the synthesis and nanostructure control of Tellurium (Te) nanotubes (NTs), and the recent progress of research into Te NTs is summarized.

In general, the work is well structured. On my opinion section 3 is the poorest section, since authors go directly to the applications, without discussing deeper the basis of the physical properties.

I include some more comments:

1) Common reader would appreciate that caption Figures will be referenced to get fastly to the original works.

Response: We agree with the reviewer that the caption Figures should be referenced and added them in all Figure captions.

Revision: Figure 1, “(a) From [8]. ©2016 Wiley-VCH Verlag GmbH & Co. KGaA, Weinheim, Germany. (b) From [10]. Copyright ©2006 American Chemical Society, Washington, United States. (c) From [42]. Copyright ©2008 American Chemical Society, Washington, United States. (d) From [11]. ©2014 Wiley-VCH Verlag GmbH & Co. KGaA, Weinheim, Germany. (e) From [10]. Copyright ©2006 American Chemical Society, Washington, United States. (f) From [12]. Copyright ©2005 American Chemical Society, Washington, United States.”

Figure 2, “From [12]. Copyright ©2014 Wiley-VCH Verlag GmbH & Co. KGaA, Weinheim, Germany.”

Figure 3, “(a) From [31]. Copyright ©2008 American Chemical Society, Washington, United States. (b-c) From [30]. Copyright ©2004 Materials Research Society, Pennsylvania, United States.”

Figure 4, “From [21]. Copyright ©2013 The Royal Society of Chemistry,Cambridge, United Kingdom.”

Figure 5, “From [7]. Copyright ©2007 WILEY-VCH Verlag GmbH & Co. KGaA, Weinheim.”

Figure 6, “From [37]. Copyright ©2004 American Chemical Society, Washington, United States.”

Figure 7, “From [38]. Copyright ©2002 WILEY-VCH Verlag GmbH & Co. KGaA, Weinheim.”

Figure 8, “From [12]. Copyright ©2005 American Chemical Society, Washington, United States.”

Figure 9, “(a-d) From [2]. Copyright ©2014 Elsevier B.V.,Amsterdam, The Netherlands. (e-f) From [1]. Copyright ©2009 American Scientific Publishers, California, United States. (g-h) From [47]. Copyright ©2013 IOP Publishing Ltd Printed in the UK & the USA.”

Figure 10, “From [52]. Copyright ©2007 WILEY-VCH Verlag GmbH & Co. KGaA, Weinheim.”

Figure 11, “From [56]. Copyright ©2021 Licensee MDPI, Basel, Switzerland.”

Figure 12, “From [59]. Copyright ©2014 Elsevier Ltd., Amsterdam, The Netherlands.”

Figure 13, “From [53]. Copyright ©2005 American Chemical Society, Washington, United States.”

2) Section 3 composed of electrical, optical and magnetoresistance is the poorest section. Authors go directly to describing specific works, without discussing the basis of their physical properties. Reader would appreciate to know them properly in this section or at least to know the best works in the state of art about this.

Response: We totally understand the reviewer’s concern, so Section 3 is carefully revised by adding more discussion of basis of their physical properties.

Revision: “Given that Te is a metalloid with relatively large spinorbit coupling [43], Te NTs are endowed with the highest electrical conductivity among inorganic elements (2×102 S·m-1), p-type narrow bandgap, and high structural rigidity over flexible NBs and NWs, they generally have highly stable electrical properties depending on their size and heteroatom doping. For another ultralong submicron Te NTs, it was found that the doped Na of only trace level could thermally scatter their weakly-bonded lattice with ease, thus enabling the resistivity of these Te NTs to decrease upon cooling (5-300 K). By employing an individual of them as the building block, a nanodevice was built through focused-ion-beam deposition to exhibit a quadratic temperature-dependent resistivity, of which the room-temperature resistivity and the ratio of 5 K resistivity/room-temperature resistivity could reach 9.854 μΩ and 0.47, respectively [40]. Te NTs with an average grain size <10 nm and wall thickness range of 15-30 nm were embedded into a field effect transistor, whose mobility was decreased to ~0.01 cm2/V·s as its phonon scattering was dominated by the Te NTs lattice that could decrease the thermal conductivity for increased thermoelectric figure of merit [23]. Besides, its field effect mobility was temperature-dependent that obeyed the Conwell-Weisskopf relationship within the temperature <250 K. For another Te NTs with easily tunable diameter (40-100 nm) by using the solvothermal method, their surface-to-volume ratio and crystallinity were optimized to fill their surface trap states and crystalline defects with more photo-generated holes. As a consequence, high photoresponsivity of 1.65×104 A·W-1 and photoconductivity gain of 5.0 ×106% were observed on the optoelectronic nanodevice based on these Te NTs [42].

3.2 Optical properties

As most Te NTs are with a single-crystalline structure, their optical properties can be modified through geometric control and chemical treatments. For example, Yu’s research group [44] reported the single-crystalline trigonal Ne NTs with sloping and hexagonal cross-sections grew along the (001) direction and had the outer diameters/wall thicknesses/lengths within 100-500 nm/50-100 nm/150-200 μm. Further, the 365 nm photoluminescence excited these Ne NTs to present blue-violet emissions (390-550 nm) for the first time, which was high related to the thickness of the nanostructures and crystallization behavior of the solvothermally reduced Te NTs [41]. Replacing the distilled water with absolute ethanol, another hydrothermally prepared Te NTs had sloping cross-sections, open ends, and relatively short lengths of 30-50 μm were also well formed. These Te NTs were found to present concentration-dependent excitation/emission, attributed to their thickness and highly anisotropic crystallization. It was also reported that the further decoration of spherical Te NPs on the shuttle-shaped Te NTs could increase their chirality considering the inherently helical chain structure with two ends to induce brand new strong red emission, beneficial for nano-optical applications [13]. Except for geometric modifications, the oxygen-related defects formed on the hexagonal column-shaped Te NTs favored the electron radiation transition from p-antibonding triple of conduction band to p-bonding triple of valence band in the latter, helping them to obtain a broad photoluminescence peak at ~532 nm [45].

3.3 Magnetoresistance properties

Ever since the positive magnetoresistance effect at low temperature was recorded on the Te microtubes by Li et al. in 2003 [32], only a few attempts have been made to explore the magnetoresistance properties of Te NTs. Later Rheem et al. [23] demonstrated that the unique magnetoresistance properties behavior could be observed on the Te NTs fabricated by the galvanic displacement, which presented a magnetoresistance ratio up to 37% (260 K), however, the related mechanism was unclear. For the latest analogues of Te NTs including layered transition metal dichalcogenides NiTe2, PdTe2 and PtTe2, irrespective of which did or did not host Weyl or Type-II Dirac fermions had high intrinsic carrier mobility, still its high purity was the prerequisite to observe maximal magnetoresistance effects [46]. In this case, the strong interconnection between carrier mobility and magnetoresistance contributed to the temperature dependence for an individual sample or the difference between the samples, via forcing carriers on Landau orbits by the applied transverse field.

3) Section 3.1: What is the reference of this assertion?

By employing an individual of them as the building block, a nanodevice was built through focused-ion-beam deposition to exhibit a quadratic temperature-dependent resistivity within 5-300 K, of which the room-temperature resistivity and the ratio of 5 K resistivity/room-temperature resistivity could reach 9.854 μΩ and 0.47, respectively.

Response: Sorry for this misleading, the reference was already cited as ref [40] (previously [38]). To avoid any confusion, the related discussion is revised as shown below.

Revision: “For another ultralong submicron Te NTs, it was found that the doped Na of only trace level could thermally scatter their weakly-bonded lattice with ease, thus enabling the resistivity of these Te NTs to decrease upon cooling (5-300 K). By employing an individual of them as the building block, a nanodevice was built through focused-ion-beam deposition to exhibit a quadratic temperature-dependent resistivity, of which the room-temperature resistivity and the ratio of 5 K resistivity/room-temperature resistivity could reach 9.854 μΩ and 0.47, respectively [40].”.

Round 2

Reviewer 1 Report

The manuscript has been sufficiently improved, and it can be considered for publication.

Reviewer 2 Report

The authors have responded perfectly to my requests.